# Tannic Acid Induces Intestinal Dysfunction and Intestinal Microbial Dysregulation in Brandt’s Voles (*Lasiopodomys brandtii*)

**DOI:** 10.3390/ani13040586

**Published:** 2023-02-07

**Authors:** Minghui Gu, Ruiyang Fan, Xin Dai, Chen Gu, Aiqin Wang, Wanhong Wei, Shengmei Yang

**Affiliations:** College of Bioscience and Biotechnology, Yangzhou University, Yangzhou 225009, China

**Keywords:** Brandt’s vole, tannic acid, intestinal function, gut microbiota

## Abstract

**Simple Summary:**

Long-term evolution has resulted in plants developing defense mechanisms against herbivores, including physical defenses (such as hard outer skin, needles, thorns, etc.) and chemical defenses (such as toxic plant secondary metabolites, etc.) that interfere with animal eating, digestion, and metabolism, thus restricting animals’ food choices. In this study, we investigated the effects of dietary tannic acid on the digestion and mucosal barrier function of Brandt’s voles. The results revealed that plant tannins are a very powerful deterrent for Brandt’s voles. This research advances our broad knowledge of interactions between plants and herbivores, as well as the ecological significance of plant secondary metabolites.

**Abstract:**

Brandt’s vole (*Lasiopodomys brandtii*) is a small herbivorous mammal that feeds on plants rich in secondary metabolites (PSMs), including tannins. However, plant defense mechanisms against herbivory by Brandt’s voles are not clearly established. This study aimed to investigate the effects of dietary tannic acid (TA) on the growth performance, intestinal morphology, digestive enzyme activities, cecal fermentation, intestinal barrier function, and gut microbiota in Brandt’s voles. The results showed that TA significantly hindered body weight gain, reduced daily food intake, changed the intestinal morphology, reduced digestive enzyme activity, and increased the serum zonulin levels (*p* < 0.05). The number of intestinal goblet and mast cells and the levels of serum cytokines and immunoglobulins (IgA, IgG, TNF-α, IL-6, and duodenal SlgA) were all reduced by TA (*p* < 0.05). Moreover, TA altered β-diversity in the colonic microbial community (*p* < 0.05). In conclusion, the results indicate that TA could damage the intestinal function of Brandt’s voles by altering their intestinal morphology, decreasing digestive ability and intestinal barrier function, and altering microbiota composition. Our study investigated the effects of natural PSMs on the intestinal function of wildlife and improved our general understanding of plant–herbivore interactions and the ecological role of PSMs.

## 1. Introduction

Herbivores require plants for survival and reproduction [1] and for maintaining a stable population size [2]. Plants have evolved systems for defense against herbivores, which hinder feeding, digestion, and metabolism, thus restricting food choices [3]. These defense mechanisms include structural features, such as needles, thorns, and spines, to prevent feeding by herbivores. Alternatively, plant secondary metabolites (PSMs) can inhibit animal growth and development [4].

The intestine is a critical site for nutrient digestion and absorption. Digestive enzymes break down the majority of the nutrients in food, and then intestinal microbes in the cecum convert undigested protein and undigested carbs to generate a wide variety of metabolites [5]. In addition, the intestine is also an important immune organ that allows the required nutrients to enter the internal environment and acts as a barrier to prevent harmful substances from penetrating the intestinal lumen [6]. The effects of PSMs on intestinal function in small mammals are incompletely understood.

Polyphenols are the most common complex secondary metabolites in plants [7]. Tannins, the most extensively studied plant polyphenols, can be classified into condensed tannins or hydrolytic tannins according to their chemical properties. Tannic acid (TA), a hydrolytic tannin, possesses antioxidant, anti-inflammatory, and antibacterial properties [8,9]. However, TA is often considered cytotoxic and may be an anti-nutritional factor for many animals and alter the growth performance of animals owing to its ability to bind to proteins [10]. Tannins have been shown to induce intestinal mucosal necrosis and villus damage in broilers [11], but gallnut TA extract decreased intestinal lesions in pigs [12]. Mbatha et al. [13] reported that TA has a negative effect on the intestinal morphology of Boer goats. Mandal et al. [14] found that TA could reduce the digestive enzyme activities in Indian major carps. Broilers fed high concentrations of TA show increased gut permeability [15]. However, little is known about the effects of TA on the intestinal function and physiology of herbivores mediated via the intestinal flora.

Brandt’s vole (*Lasiopodomys brandtii*) is a non-hibernating herbivore, largely inhabiting Inner Mongolia and other regions with degraded pastures [16]. It feeds on grassland vegetation and is therefore extremely destructive to grasslands. *Leymus chinensis*, the dominant plant in Inner Mongolia grassland, the preferred food of voles, contains TA [17]. Accordingly, Brandt’s vole is a suitable subject to explore the effects of PSMs on the survival and reproduction of herbivores. TA can affect ovarian development in Brandt’s voles, thus affecting their reproductive capacity [18]. However, it is not clear whether TA affects intestinal function, and therefore the survival of the species.

In the present study, we examined the impact of TA (at doses not exceeding the average content in the typical diet) on Brandt’s voles in terms of (1) growth performance, (2) intestinal morphology, (3) the activity of digestive enzymes and cecal fermentation, and (4) immune and microbial barrier functions of the intestine. These analyses allowed us to gauge the effects of TA on the intestinal function of Brandt’s voles, thereby potentially providing a new theoretical basis for the understanding of plant–herbivore interactions and the ecological role of PSMs.

## 2. Materials and Methods

### 2.1. Animal Care, Treatments, and Sample Collection

Twenty-four male 13-week-old Brandt’s voles, weighing 52.16 ± 0.84 g, were used in this study. Due to the intricate fluctuations in sex hormone levels in female Brandt’s voles over the reproductive cycle, adult male voles were used for this experiment [19]. The voles were bred under controlled environmental conditions (22–25 °C, 50 ± 5% relative humidity). The photoperiod was 16L:8D. After weaning, voles were housed individually in battery cages and given free access to food and water throughout the experiment. The Faculty of Veterinary Medicine at Yangzhou University’s Animal Care and Use Committee gave its approval for all procedures (No. NSFC2020-SKXY-6).

Based on the average concentration of TA in *Leymus chinensis* and field consumption by Brandt’s voles, a wild vole weighing 50 g can consume 0.03–0.069 g of TA daily [18]. Brandt’s voles were given 0.2 mL TA by gavage every day. Three groups of eight voles each were given 0 (control), 600 (LT), or 1200 mg kg^−1^ d^−1^ (HT) TA over the course of 18 days. The gavage was given between 09:00 and 10:00 am. Body weight (BW) and food intake were measured every 3 days. At the end of the experiment, the blood was collected in an anticoagulant tube treated with heparin sodium through eyeball extirpation. Serum samples were obtained after centrifugation at 3000× *g* at 4 °C for 30 min and maintained at −20 °C until further analysis. The viscera, including the liver, kidney, heart, and spleen, were weighed. The duodenum, jejunum, and ileum were isolated and rinsed with saline several times and then fixed in 4% paraformaldehyde for histological examination. The remaining segments of the small intestine, cecal contents, and colonic contents were collected in sterile centrifuge tubes and stored at −80 °C until further analyses.

### 2.2. Intestinal Morphology

In a 25% sucrose solution at 4 °C, the fixed intestine segments were first dehydrated to the bottom before being moved to a 30% sucrose solution to finish the process. The samples were subsequently sectioned at a thickness of 20 μm using a freezing microtome (CM1100-1; Leica, Nussloch, Germany) at −20 °C and then stained with hematoxylin and eosin (H&E) as previously described [20]. The villus height (VH) and crypt depth (CD) were measured under a microscope (U-CBS; Olympus, Tokyo, Japan) at 200× magnification. Fixed tissues were stained with periodic acid–Schiff (PAS) solution to detect goblet cells, and mast cells were stained with toluidine blue (TB) and visualized and counted by light microscopy as in an earlier study [21]. Five sections were taken from each small intestine of each vole for statistical analysis.

### 2.3. Intestinal Digestive Enzyme Assays

The small intestinal segments and the cecum contents were thawed and homogenized in ice-cold 0.9% physiological saline solution. The homogenate was then centrifuged at 4000× *g* for 10 min at 4 °C. The supernatant was analyzed for the activity of digestive enzymes, including maltase, invertase, lipase, cellulase, and amylase. Activity levels were measured using a microplate reader (BioTek Instruments Co., Ltd., Winooski, VT, USA) according to the commercial test kits (Nanjing Jiancheng Bioengineering Institute, Nanjing, China).

### 2.4. Ammonia Nitrogen and Short-Chain Fatty Acids (SCFA) Contents

For the analysis of SCFAs (mainly acetate, propionate, and butyrate), a 0.2 g cecal content was dissolved in 1 mL of sterile water and centrifuged at 10,000× *g* for 15 min at 4 °C. The supernatant was added to metaphosphoric acid at a ratio of 9:1 and placed on ice for 2 h. The mixture was then centrifuged at 12,000× *g* for 10 min at 4 °C, and the supernatant passed through a 0.22 mm nylon membrane filter [22]. Then, 988 µL of distilled water, 10 µL of acetate, 1 µL of propionate, and 1 µL of butyrate were mixed together to create standards. Finally, the solution was injected into a gas chromatographic system (Agilent 7080A; Santa Clara, CA, USA) to determine SCFAs concentrations. The peak area of each SCFA in the standards and the samples was determined using an Agilent Chemstation. The ammonia nitrogen (NH_3_-N) content was measured using a microplate reader (BioTek Instruments Co., Ltd.), according to an earlier study [23].

### 2.5. Determination of Intestinal Permeability, and Immunoglobulin and Cytokine Levels

Plasma endotoxins, zonulin, immunoglobulin subsets (IgA and IgG), and cytokines (TNF-α and IL-6) were measured using ELISA kits (Jianglai Bioengineering Institute, Shanghai, China) according to the manufacturer’s instructions and as previously reported [24]. Approximately 0.1 g of duodenal, jejunal, and ileal mucosal samples were weighed, homogenized with cold saline, and centrifuged at 4 °C for 10 min. The levels of secretory immunoglobulin A (SIgA) were determined using kits (Jianglai Bioengineering Institute) according to the instructions. The intra-group and inter-group coefficients of variation for all indicators were <9% and <13%, respectively.

### 2.6. 16S rRNA High-Throughput Gene Sequencing

Taking the cost into consideration, we randomly selected six of the eight voles in each group for 16S rRNA high-throughput gene sequencing. DNA was extracted from colonic fecal samples using the Stool DNA Kit (Tiangen Biotech (Beijing) Co., Ltd., Beijing, China) [22]. Following amplification, PCR products were separated by electrophoresis on a 1.8% agarose gel to check for the integrity, and the NanoDrop 2000 (Thermo Scientific, Waltham, MA, USA) was used to gauge the quantity and quality of DNA. Using total DNA as a template, the V3–V4 region of 16S rRNA was amplified by PCR and high-throughput sequencing analysis was performed on the Illumina NovaSeq 6000 platform by Beijing Biomarker Technologies Co., Ltd. (Beijing, China) [25]. Ace, Chao1, Shannon, and Simpson indices were calculated at the operational taxonomic unit (OUT) level to evaluate α-diversity. A principal coordinates analysis (PCoA) was performed using the “ape” package of R (version 4.0.4) and ANOSIM was performed using the “vegan” package of R (version 4.0.4) to evaluate β-diversity [19]. The online LEfSe program available on the Huttenhower Lab server was used to identify significant biomarkers for the three groups [26]. The default setting for the LDA (linear discriminant analysis) score was 2. The prediction and annotation of biological functions of colonic bacterial communities were performed using the Kyoto Encyclopedia of Genes and Genomes (KEGG) pathway database and PICRUSt [27]. The NCBI’s Sequence Read Archive has received the 16S sequencing dataset with the Bioproject ID PRJNA898866.

### 2.7. Statistical Analysis

The body weight change of voles was studied using a repeated measures analysis, followed by Tukey’s post hoc tests. Permutational multivariate analysis of variance (PERMANOVA) with Bray–Curtis distance matrices was performed using the nested adonis function of the “vegan” package in R (version 4.0.4) to compare β-diversity estimates of the cecal microbial community among groups. We compared OTU biomarkers and pathway enrichment using the conventional non-parametric Kruskal–Wallis test. Other data were analyzed by one-way analysis of variance (ANOVA) followed by Tukey’s post hoc test using SPSS (Version 26; SPSS Inc., Chicago, IL, USA). Data are presented as the mean ± SEM. Values of *p* < 0.05 were considered statistically significant.

## 3. Results

### 3.1. Effect of TA on Growth Performance in Brandt’s Voles

There were no significant differences in the initial BW of voles between the three groups (*p* > 0.05, Table 1). Growth curves during the TA intragastric administration period (18 days) are shown in Figure 1. The BW changes in the LT and HT group were less than those in the control group at all time points (*p* = 0.01). Average daily food intake (ADFI) was significantly lower in the LT and HT group than in the control group (*p* = 0.007 and *p* = 0.002, respectively, Table 1). No differences were observed in the liver, kidney, heart, and spleen indices or in the intestinal length between the control and TA groups (*p* > 0.05, Table 1).

### 3.2. Effect of TA on the Intestinal Morphology of Brandt’s Voles

As shown in Table 2, VH was significantly lower in the TA treatment groups than in control group in the duodenum, jejunum, and ileum (*p* = 0.015, *p* = 0.040, and *p* = 0.015, respectively). Moreover, the VH/CD ratios of the duodenum, jejunum, and ileum were markedly lower in the TA group than in the control group (*p* = 0.007, *p* = 0.009, and *p* = 0.006, respectively). No significant differences were observed in the CD of the duodenum, jejunum, and ileum among the three groups (*p* > 0.05).

### 3.3. Effect of TA on Intestinal Enzyme Activities of Brandt’s Voles

As shown in Table 3, HT significantly decreased amylase activity in the jejunum (*p* = 0.044) and cecum (*p* = 0.034) and lipase activity in the duodenum (*p* = 0.031). Amylase activity in the ileum was significantly lower in the LT group than in the control group (*p* = 0.048). Cellulase activity was significantly lower (*p* = 0.043 and *p* = 0.046, respectively) in the LT and HT groups than in the control group. However, no marked differences were observed in the maltase and invertase activities among the three groups (*p* > 0.05).

### 3.4. Effect of TA on Cecal Fermentation of Brandt’s Voles

As shown in Table 4, a low dosage of TA resulted in a significantly lower acetic acid (*p* = 0.036) concentration than that of the control group. Voles in the HT group had significantly lower concentrations of butyric acid (*p* = 0.023) than those in the control group. However, no marked differences were observed in the propionic acid content, NH_3_-N concentration, and ruminal pH among the three groups (*p* > 0.05).

### 3.5. Effect of TA on Intestinal Barrier Function of Brandt’s Voles

#### 3.5.1. Effect of TA on Intestinal Permeability of Brandt’s Voles

As shown in Figure 2A,B, the levels of zonulin in the LT and HT groups were significantly higher than those in the control group (*p* = 0.033 and *p* = 0.015, respectively). No marked differences were observed in endotoxin levels among the three groups (*p* > 0.05).

#### 3.5.2. Effect of TA on Cytokine and Immunoglobulin Levels in Brandt’s Voles

As shown in Figure 3A–E, a high dosage of TA significantly decreased the levels of serum IgA, IgG, TNF-α, and IL-6 (*p* = 0.010; *p* = 0.008; *p* = 0.034; and *p* = 0.044, respectively) compared with those in the control group. HT significantly decreased the concentration of SIgA in the duodenum compared with that of the control group (*p* = 0.014). TA had no effect on the jejunal SIgA concentration, whereas the ileal SIgA concentrations were increased in voles gavaged with LT (*p* = 0.010).

#### 3.5.3. Effect of TA on Goblet Cell and Mast Cell Counts in the Small Intestine of Brandt’s Voles

As shown in Figure 4, there were significantly fewer goblet cells in ileal sections in the LT and HT groups than that in the control group (*p* = 0.004 and *p* = 0.010, respectively). The mast cell counts in the duodenum and jejunum were significantly lower in the HT group than that in the control group (*p* = 0.016 and *p* = 0.026, respectively).

#### 3.5.4. Effect of TA on the Gut Microbiota in the Colon of Brandt’s Voles

A total of 18 samples were sequenced using paired-end technology, yielding 1,441,241 reads. After quality control and paired-end read splicing, 1,436,831 clean reads were produced in total, with at least 79,493 clean reads produced for each sample.

Microbial communities of different groups at different levels are shown in Appendix A. The most abundant bacterial phyla were Firmicutes, Bacteroidetes, and Desulfobacterota. The most abundant bacterial orders were Lachnospirales, Bacteroidales, and Oscillospirales. The most abundant bacterial classes were Clostridia, Bacteroidia, and Desulfovibrionia. The most abundant families were Lachnospiraceae, Muribaculaceae, and Ruminococcaceae. The most abundant genera were unclassified_*Lachnospiraceae*, unclassified_*Muribaculaceae,* and *Lachnospiraceae_NK4A136_group*.

The colonic microbial α-diversity estimates for Brandt’s voles in each group are shown in Table 5. As determined using the Kruskal–Wallis test, there were no significant differences in the Ace, Chao1, Simpson, and Shannon indices among the groups, indicating that TA did not alter the colonic microbial α-diversity (*p* > 0.05). PERMANOVA showed that the β-diversity of the colonic microbial community changed significantly 18 days after TA administration (*F* = 1.293, *p* = 0.050), with a significant difference between the HT group and control group (*p* = 0.048). In addition, a PCoA with Bray–Curtis dissimilarity metrics was used to compare the differences in colon microbiome structure of voles after TA gavage. As shown in Figure 3A,B, PCoA1 and PCoA2 explained 17.38% and 11.67% of the total variance, respectively. The ANOSIM results showed that the structural difference in the colonic microbiome among the three groups was less than that of the HT group (*R* = 0.108, *p* = 0.048).

LEfSe (LDA effect size) was used to identify biomarkers with significant differences among the three groups (score > two). As shown in Figure 5C,D and Figure 6, the relative abundance of the phylum Firmicutes was significantly lower after 18 days of intragastric administration of HT than those in the control and LT groups (*p* = 0.020 and *p* = 0.027, respectively). HT also significantly altered the relative abundance of the order Monoglobales compared with the LT group (*p* = 0.017), and altered the relative abundance of order Sphingomonadales compared with the control group and LT group (*p* < 0.001 and *p* = 0.031, respectively). The relative abundance of families Oxalobacteraceae, Sphingomonadaceae, and Butyricicoccaceae were significantly lower in the HT group than in the control group (*p* = 0.023; *p* = 0.001; and *p* = 0.007, respectively). HT significantly increased the relative abundance of the families uncultured_rumen_bacterium and Defluviitaleaceae compared with those in the control group (*p* = 0.015 and *p* = 0.012, respectively). LT significantly increased the relative abundance of families Monoglobaceae and Butyricicoccaceae compared with those in the HT group (*p* = 0.017 and *p* = 0.008, respectively). At the genus level, HT significantly decreased the relative abundances of *Roseburia*, *Papillibacter*, *Oxalobacter*, *UCG_009*, and *Sphingomonas* compared with those in the control group (*p* = 0.02; *p* = 0.012; *p* = 0.023; *p* = 0.006; and *p* < 0.001, respectively). HT significantly decreased the relative abundances of *Monoglobus*, *UCG_009*, *Spingomonas*, and *Lachnospiraceae_FCS020_group* compared with those in the LT group (*p* = 0.017; *p* = 0.009; *p* = 0.031; and *p* = 0.005, respectively). The relative abundances of *uncultured_rumen_bacterium*, *[Eubacterium]_nodatum_group*, and *Defluviitaleaceae_UCG_011* were significantly higher in the HT group than in the control group (*p* = 0.013; *p* = 0.031; and *p* = 0.012, respectively). The relative abundances of *uncultured_rumen_bacterium* and *[Eubacterium]_nodatum_group* were significantly higher in the HT group than in the LT group (*p* = 0.027 and *p* = 0.007, respectively).

As shown in Table 6 and Appendix A, TA altered the function of 18 KEGG pathways in the colonic bacterial communities of Brandt’s voles. Ubiquinone and other terpenoid-quinone biosynthesis, steroid hormone biosynthesis, and Glycosphingolipid biosynthesis—ganglio series, lysosome, apoptosis, adipocytokine signaling pathway, pertussis, and Staphylococcus aureus infection were more dominant in the colonic bacterial community in the HT groups than in the control and LT groups (*p* = 0.017, 0.031; *p* = 0.027, 0.020; *p* = 0.023, 0.035; *p* = 0.020, 0.004; *p* = 0.006, 0.023; *p* = 0.004, 0.023; *p* = 0.017, 0.031; and *p* = 0.007, 0.020; respectively). Other glycan degradation, Lipopolysaccharide biosynthesis, biosynthesis of siderophore group nonribosomal peptides, PPAR signaling pathway, and apoptosis—fly, protein digestion, and absorption were more dominant in the colonic bacterial community in the HT groups than in the control groups (*p* = 0.008; *p* = 0.005; *p* = 0.008; *p* = 0.011; *p* = 0.013; *p* = 0.007; and *p* = 0.015, respectively). Bacterial chemotaxis, flagellar assembly, and Salmonella infection were more dominant in the colonic bacterial community in the control groups than in the HT groups (*p* = 0.008; *p* = 0.006; and *p* = 0.008, respectively).

## 4. Discussion

Previous research on tannins has focused more on their effects on the growth performance of poultry animals, and less on that of wildlife. Hydrolytic tannins are easily hydrolyzed into glucose and gallic acid and can be subdivided into two classes, gallotannins and ellagitannins [28]. In this study, we investigated the effects of TA, a type of gallotannin, on the growth performance, intestinal morphology, digestive enzyme activity, cecal fermentation, intestinal barrier function, and gut microbiota composition of Brandt’s voles.

Previous studies have shown that tannins can affect the growth performance of animals. Tong et al. [29] showed that plant tannins can improve the growth performance of broilers. By contrast, Choi et al. [15] reported that the astringency and bitter taste of TA–saliva protein complexes reduced feed intake, and thus, reduce broilers growth performance. Consistent with the study of Choi et al., we found that TA reduced the BW and ADFI of voles compared with those of the control group, indicating that TA has an adverse effect on growth performance. The differences between studies might be owing to the differences in the source and amount of tannins, features of experimental animals, diet and living environment, duration of the experiments, and potential correlations between these factors.

Proper functioning of the intestine depends on its morphological integrity. A change in intestinal morphology can affect the digestion, absorption, and transportation of nutrients [30]. VH, CD, and the VH/CD ratio are the main indices used to evaluate the morphological structure of the small intestine [31]. TA has been found to cause significant changes in these intestinal parameters. Mbatha et al. [13] reported that TA had a negative effect on intestinal morphology by damaging the gut villi of Boer goats. Consistent with previous results, we did not detect significant differences in the length of intestinal segments when voles were fed TA at varying concentrations. However, TA feeding altered the intestinal morphology of voles by decreasing VH and the VH/CD ratio in the intestine. The current results demonstrated that TA damages the structure of the intestinal mucosa of Brandt’s voles.

In vitro digestive enzyme activity levels directly affect the absorption and utilization of nutrients, thus affecting the health status and growth rate of animals. The effects of TA on digestive enzyme activities have not been thoroughly studied. Studies have shown that TA can reduce the enzymatic activities of lipase, amylase, and protease in Indian major carps [14]. In this study, TA decreased amylase, cellulase, and lipase activities in different intestinal segments. TA binds to proteins, thereby reducing their absorption. After protein binding, a sufficient hydrophobic layer is formed on the surface, which is affected by features of the protein (size, conformation, and charge), as well as the number and stereospecificity of binding sites on polyphenols and protein molecules [32]. The effects of TA on digestive enzymes differ between segments of the small intestine, which may reflect functional differences among segments.

Most nutrients in food are digested and absorbed in the stomach and small intestine. Undigested protein and undigestible carbohydrates in the cecum are metabolized to generate a large number of metabolites by intestinal microorganisms [33]. Brandt’s voles exhibit post-gut fermentation [34]. The cecal pH, NH_3_-N concentration, and SCFA content are important indices used to evaluate the level of cecal fermentation [35]. It is generally accepted that SCFAs (mainly including acetate, propionate, and butyrate) play an important role as an energy source [36]. Ammonia nitrogen is a major protein catabolite and an important substrate for bacterial protein synthesis by cecal microorganisms [37]. Biagi et al. [38] reported that TA significantly reduced the concentration of ammonia, isobutyric acid, and isovaleric acid in the cecum of weaned piglets. In line with previous findings, TA supplementation decreased the acetic acid and butyric acid contents in the cecum, indicating that fermentation decreased in TA-fed Brandt’s voles, compared with those in the control group voles.

The levels of endotoxins and zonulin in plasma are indicators of intestinal permeability [21]. Increased intestinal permeability allows toxins to enter the blood and intestines, where they cause the production of several cellular inflammatory factors that result in intestinal inflammation [39]. Choi et al. [15] found that broilers fed high concentrations of TA displayed increased gut permeability. In this study, we found that TA increased plasma levels of zonulin compared with that in the control group, implying that TA could increase intestinal permeability.

The distribution and function of intestinal goblet cells play an essential role in maintaining an intact gut tissue morphology [40]. The intestinal mucosal barrier formed by mucus secreted by goblet cells can prevent exogenous pathogenic substances from entering intestinal epithelial cells and protect the intestinal tissue morphology [41]. In the present study, TA reduced the number of goblet cells in the ileum of voles, indicating that TA had adverse effects on the intestinal chemical barrier in the vole ileum. Additionally, the number of small intestinal goblet cells tended to increase from the duodenum to the ileum, consistent with the results of a previous study [42]. Mast cells exhibit immune activity and secrete a variety of inflammatory mediators, such as TNF-α, in the intestine [43]. In this study, a high concentration of TA reduced the number of mast cells in the duodenum and jejunum of voles.

The immune function of the intestinal mucosa is mediated by immune cells and cytokines [44]. Immunoglobulins, especially IgG and IgA, play an important role in humoral immunity. An increase in the secretion of immunoglobulins indicates an improvement in immune function. SIgA is a major component of the intestinal mucosal defense system and a barrier to maintain the stability of the intestinal mucosal internal environment. Jafari et al. [45] noted that oak acorn containing high levels of tannins can reduce immunoglobulin synthesis. In the present study, HT treatment reduced the levels of TNF-α, IL-6, lgG, and lgA in the serum and SlgA in the duodenum of voles, while LT treatment increased the level of SlgA in the ileum of voles. These findings suggest that TA had negative effects on immune function in voles.

In addition to immune barriers, commensal microorganisms in the gut form part of the intestinal epithelial barrier. These microbes interact with the host, forming a coevolutionary relationship [46]. In this study, TA did not change the microbial alpha diversity; however, HT significantly changed the β-diversity of the colonic microbial community compared with that of the control group. Furthermore, an enrichment in biomarker OTUs was observed. These findings indicate that TA could alter the structure of the colonic microbiota. In the present study, the dominant bacterial phyla were Firmicutes and Bacteroidetes, consistent with previous results [6]. Firmicutes can convert cellulose into volatile fatty acids to promote fiber degradation, growth, and weight gain [47]. Bacteroidetes effectively break down carbohydrates, preventing obesity [48]. In our study, we found a significantly lower relative abundance of Firmicutes and a lower Firmicutes/Bacteroidetes ratio, a marker of gut dysbiosis, in the HT group than in the control and LT groups. This result is consistent with the observed changes in BW, suggesting that weight loss is related to the HT treatment-induced reduction in Firmicutes.

Some bacteria, such as *Roseburia*, Lachnospiraceae, and Butyricicoccaceae, produce SCFAs [49]. *Roseburia* have been reported to produce SCFAs that break down indigestible carbohydrates. These compounds are usually involved in energy production and can protect the gut from pathogens [50]. Furthermore, *Roseburia* may play an important role in controlling inflammatory processes, especially intestinal inflammation. The dysregulation of *Roseburia* may affect a variety of metabolic pathways [51]. Lachnospiraceae, a potentially beneficial bacterium in the gut, produces acetic acid and butyrate for host energy and the maintenance of gut barrier integrity [52]. Butyricicoccaceae can also produce butyrate and are related to ulcerative colitis [53]. In the present study, the relative abundance of f_Butyricicoccaceae, g_*Roseburia*, and g_*Lachnospiraceae* was significantly lower in the HT group than in the control and LT groups. Furthermore, HT significantly reduced the butyrate content. Thus, we speculate that TA can decrease the abundance of these bacteria to reduce the production of SCFAs, thus damaging overall health. *Oxalobacter* is a key oxalate-degrading bacterium in the mammalian gut. The degradation of oxalate in the intestine plays a key role in preventing nephrotoxicity in animals that feed on oxalate-rich plants [54]. *Sphingomonas* can degrade macromolecular organic matter and is antagonistic to some pathogens. It can promote growth [55]. *Papillibacter* are also thought to have probiotic effects [56]. In our study, we found that HT treatment significantly reduced the relative abundance of o_Sphingomonadales, f_Oxalobacter, f_Sphingomonadaceae, g_*Papillibacter*, and g_*Sphingomonas* compared with the control and LT groups. Furthermore, the relative abundance of some pathogenic bacteria, such as f_Defluviitaleaceae and g_*[Eubacterium]_nodatum_group*, was higher in the HT group than in the control group. Therefore, HT reduced the abundance of probiotics and increased the abundance of pathogenic bacteria in Brandt’s voles. KEGG function analysis revealed that TA induced a change in the metabolism, cellular processes, and organismal systems in the colonic bacteria and that the change was dose dependent. These findings provide a basis for further studies of the role of the gut microbiota in the health, environmental adaptation, and metabolism of Brandt’s voles.

## 5. Conclusions

In summary, our results demonstrated that TA, especially at high concentrations, can influence intestinal function in Brandt’s voles by altering their intestinal morphology, decreasing digestive enzyme activity, attenuating their cecal fermentation, decreasing intestinal immune function, and altering the microbiota structure. Thus, we speculate that plant tannins are a highly effective defense against Brandt’s voles at both the individual and population levels.

## Figures and Tables

**Figure 1 animals-13-00586-f001:**
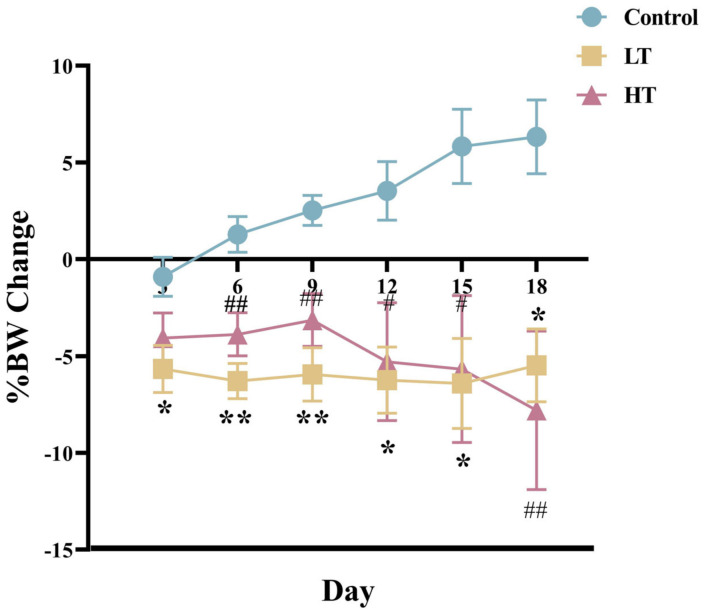
Body weight change during the 18-day TA intragastric administration to male Brandt’s voles. Control, 0 mg•kg^−1^d^−1^ TA; LT, 600 mg•kg^−1^d^−1^ TA; HT, 1200 mg•kg^−1^d^−1^ TA; BW, body weight. %BW change = (Final BW − Initial BW)/Final BW × 100%. Data are presented as mean ± SEM, *n* = 8. * *p* < 0.05, ** *p* < 0.01 LT vs. control group; ^#^
*p* < 0.05, ^##^
*p* < 0.01 HT vs. control group.

**Figure 2 animals-13-00586-f002:**
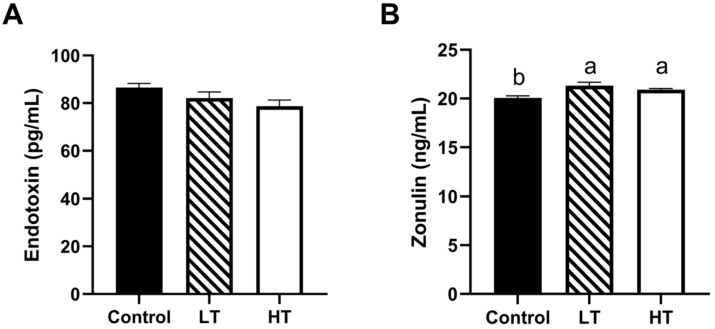
Effect of TA on the intestinal permeability of Brandt’s voles. Plasma levels of endotoxin (**A**) and zonulin (**B**). Control, 0 mg•kg^−1^d^−1^ TA; LT, 600 mg•kg^−1^d^−1^ TA; HT, 1200 mg•kg^−1^d^−1^ TA; ^a, b^ Means with different letters differ significantly (*p* < 0.05). Data are presented as mean ± SEM, *n* = 8.

**Figure 3 animals-13-00586-f003:**
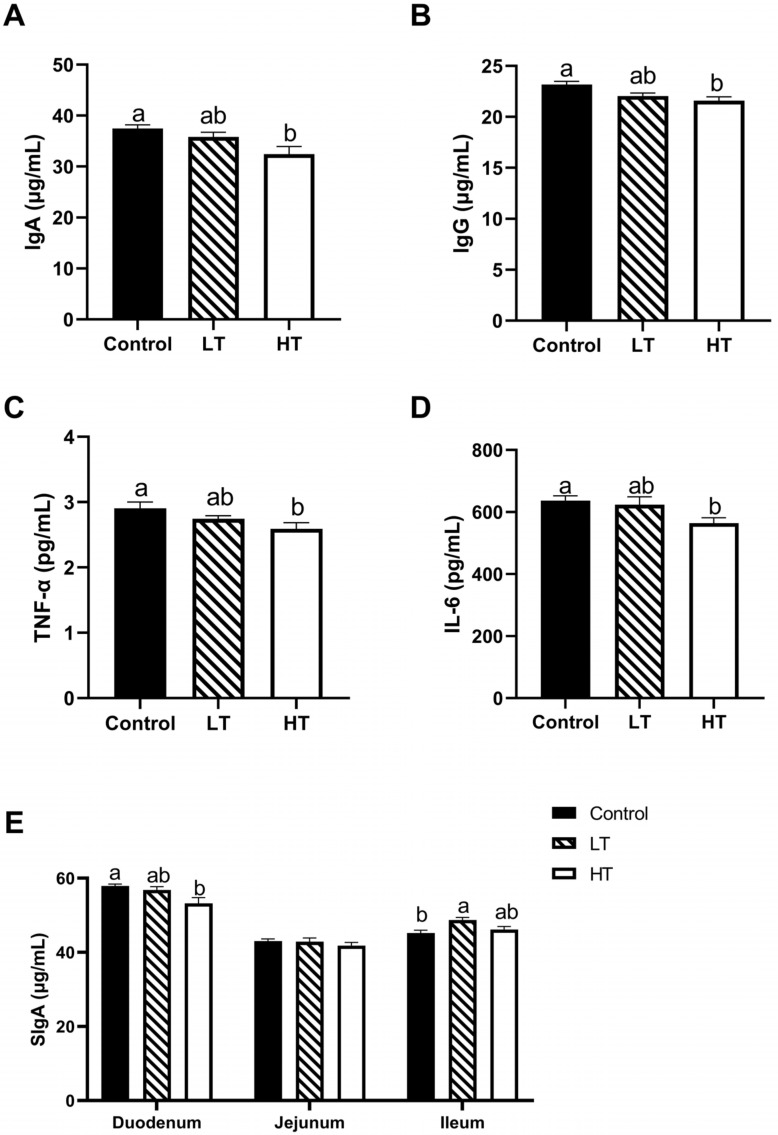
Effect of TA on cytokine and immunoglobulin of Brandt’s voles. Plasma levels of lgA (**A**), lgG (**B**), TNF-α (**C**), and IL-6 (**D**). SlgA levels of duodenum, jejunum, and ileum (**E**). Control, 0 mg•kg^−1^d^−1^ TA; LT, 600 mg•kg^−1^d^−1^ TA; HT, 1200 mg•kg^−1^d^−1^ TA; lgA, immunoglobulin A; lgG, immunoglobulin G; TNF-α, tumor necrosis factor-α; IL-6, interleukin-6; SlgA, secretory immunoglobulin A. ^a, b^ Means with different letters differ significantly (*p* < 0.05). Data are presented as mean ± SEM, *n* = 8.

**Figure 4 animals-13-00586-f004:**
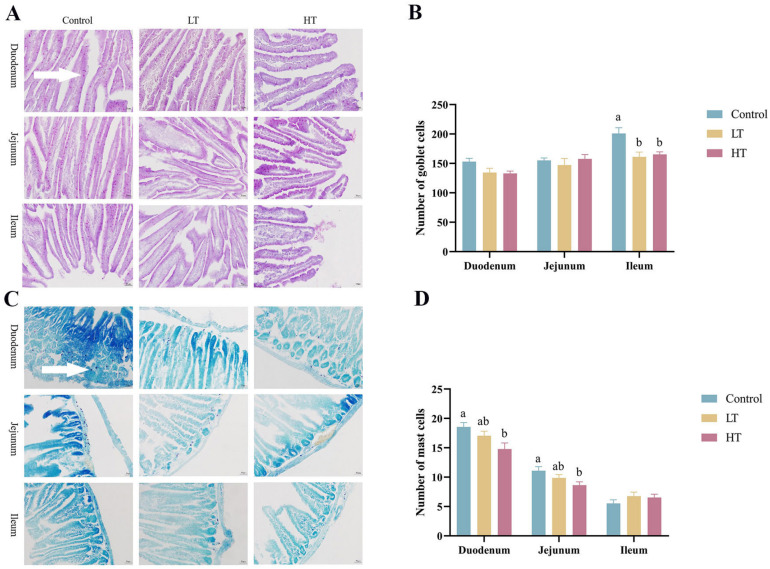
Effect of TA on goblet cell and mast cell counts in small intestine of Brandt’s voles. (**A**) Representative images of PAS-stained histological samples of the small intestine, scale bar = 50 µm. The white arrow points to goblet cell. (**B**) Number of goblet cells. (**C**) Representative images of TB-stained histological samples of the small intestine, scale bar = 50 µm. The white arrow points to mast cell. (**D**) Number of mast cells. Control, 0 mg•kg^−1^d^−1^ TA; LT, 600 mg•kg^−1^d^−1^ TA; HT, 1200 mg•kg^−1^d^−1^ TA. ^a, b^ Means with different letters differ significantly (*p* < 0.05). Data are presented as mean ± SEM, *n* = 8.

**Figure 5 animals-13-00586-f005:**
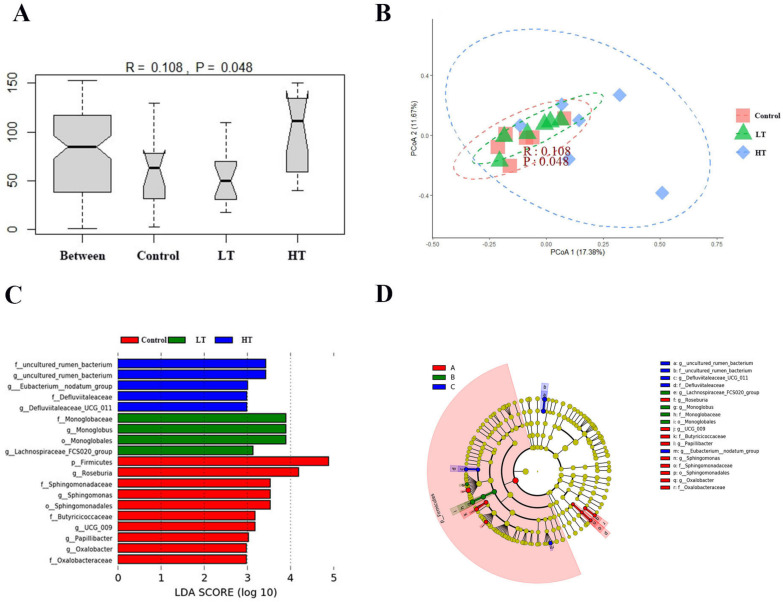
Effect of TA on the colonic microbial β-diversity and colonic microbial biomarker abundance of Brandt’s voles. (**A**) ANOSIM analysis plot. (**B**) Principal coordinates analysis. (**C**) LDA distribution. (**D**) Cladogram. The threshold of LDA score was 2.0. Control, 0 mg•kg^−1^d^−1^ TA; LT, 600 mg•kg^−1^d^−1^ TA; HT, 1200 mg•kg^−1^d^−1^ TA.

**Figure 6 animals-13-00586-f006:**
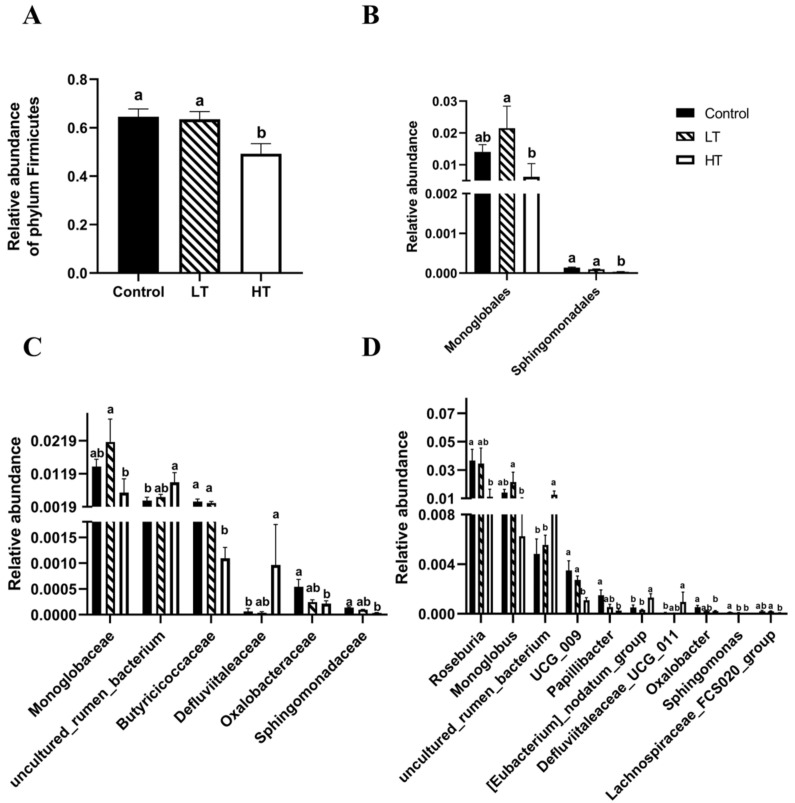
Effect of TA on the relative abundance of colonic microbial taxa of Brandt’s voles. (**A**) phylum level; (**B**) order level; (**C**) family level; (**D**) genus level. Control, 0 mg•kg^−1^d^−1^ TA; LT, 600 mg•kg^−1^d^−1^ TA; HT, 1200 mg•kg^−1^d^−1^ TA. ^a, b^ Means with different letters differ significantly (*p* < 0.05). Data are presented as mean ± SEM, *n* = 6.

**Table 1 animals-13-00586-t001:** Effect of dietary TA supplementation on growth performance of male Brandt’s voles.

	Control	LT	HT
Initial BW (g)	52.10 ± 1.58	51.69 ± 1.49	52.70 ± 1.48
Final BW (g)	55.35 ± 1.71	48.82 ± 1.51	48.66 ± 1.48
ADFI (g)	7.31 ± 0.64 ^a^	5.02 ± 0.27 ^b^	4.58 ± 0.45 ^b^
Organ index (%)			
Liver	3.12 ± 0.12	3.34 ± 0.10	3.28 ± 0.12
Spleen	0.07 ± 0.00	0.08 ± 0.00	0.07 ± 0.00
Kidney	0.83 ± 0.03	0.83 ± 0.02	0.89 ± 0.05
Heart	0.52 ± 0.02	0.53 ± 0.02	0.47 ± 0.02
Intestinal length (cm)			
Small intestine	26.19 ± 0.21	24.50 ± 0.75	24.44 ± 0.72
Cecum	9.00 ± 0.16	9.50 ± 0.49	9.42 ± 0.53
Colon	19.56 ± 0.53	19.25 ± 0.40	19.06 ± 0.59
Total intestine	54.75 ± 0.53	53.25 ± 0.94	53.25 ± 1.53

^a, b^ Means in the same row with different letters differ significantly (*p* < 0.05). Control, 0 mg•kg^−1^d^−1^ TA; LT, 600 mg•kg^−1^d^−1^ TA; HT, 1200 mg•kg^−1^d^−1^ TA; BW, body weight; ADFI, average daily food intake. Data are presented as mean ± SEM, *n* = 8.

**Table 2 animals-13-00586-t002:** Effect of TA on the intestinal morphology of Brandt’s voles.

	Control	LT	HT
Duodenum			
Villus Height (μm)	865.78 ± 30.16 ^a^	749.46 ± 30.04 ^b^	755.73 ± 25.65 ^b^
Crypt Depth (μm)	134.98 ± 5.41	143.86 ± 4.86	150.54 ± 6.86
VH/CD	6.47 ± 0.31 ^a^	5.26 ± 0.32 ^b^	5.09 ± 0.27 ^b^
Jejunum			
Villus Height (μm)	770.75 ± 43.86 ^a^	652.28 ± 38.87 ^ab^	609.85 ± 45.95 ^b^
Crypt Depth (μm)	107.17 ± 6.41	116.69 ± 7.05	114.41 ± 6.25
VH/CD	7.36 ± 0.57 ^a^	5.66 ± 0.31 ^b^	5.37 ± 0.38 ^b^
Ileum			
Villus Height (μm)	500.62 ± 19.14 ^a^	428.47 ± 17.15 ^b^	427.42 ± 18.92 ^b^
Crypt Depth (μm)	91.43 ± 6.29	93.41 ± 6.24	96.54 ± 3.79
VH/CD	5.58 ± 0.28 ^a^	4.68 ± 0.28 ^b^	4.42 ± 0.08 ^b^

^a, b^ Means in the same row with different letters differ significantly (*p* < 0.05). Control, 0 mg•kg^−1^d^−1^ TA; LT, 600 mg•kg^−1^d^−1^ TA; HT, 1200 mg•kg^−1^d^−1^ TA; VH, Villus Height; CD, Crypt Depth. Data are presented as mean ± SEM, *n* = 8.

**Table 3 animals-13-00586-t003:** Effect of TA on the intestinal enzyme activities of Brandt’s voles.

	Control	LT	HT
Duodenum			
Amylase (U/mg)	1.35 ± 0.72	1.31 ± 0.12	1.180 ± 0.18
Maltase (U/mg)	1897.61 ± 108.61	1763.21 ± 123.27	1805.33 ± 75.03
Invertase (U/mg)	499.00 ± 43.61	458.02 ± 43.77	432.04 ± 22.68
Lipase (U/g)	14.68 ± 1.23 ^a^	11.82 ± 1.30 ^ab^	10.41 ± 0.63 ^b^
Jejunum			
Amylase (U/mg)	0.70 ± 0.06 ^a^	0.46 ± 0.08 ^ab^	0.43 ± 0.08 ^b^
Maltase (U/mg)	1619.76 ± 111.69	2067.78 ± 204.82	2005.52 ± 250.81
Invertase (U/mg)	449.41 ± 47.69	579.95 ± 69.46	557.80 ± 75.17
Lipase (U/g)	3.30 ± 0.44	3.77 ± 0.35	2.91 ± 0.35
Ileum			
Amylase (U/mg)	0.47 ± 0.08 ^a^	0.28 ± 0.03 ^b^	0.30 ± 0.03 ^ab^
Maltase (U/mg)	2211.54 ± 168.28	2149.43 ± 199.97	2556.25 ± 206.96
Invertase (U/mg)	605.16 ± 61.67	549.96 ± 60.17	714.02 ± 91.02
Lipase (U/g)	2.76 ± 0.14	2.24 ± 0.20	2.23 ± 0.23
Cecum			
Cellulase (U/mg)	1274.42 ± 15.85 ^a^	1195.12 ± 20.65 ^b^	1195.79 ± 27.03 ^b^
Amylase (U/mg)	6.43 ± 0.35 ^a^	5.89 ± 0.25 ^ab^	5.02 ± 0.47 ^b^
Protease (U/mg)	10.38 ± 1.30	9.86 ± 0.69	11.23 ± 1.16

^a, b^ Means in the same row with different letters differ significantly (*p* < 0.05). Control, 0 mg•kg^−1^d^−1^ TA; LT, 600 mg•kg^−1^d^−1^ TA; HT, 1200 mg•kg^−1^d^−1^ TA; data are presented as mean ± SEM, *n* = 8.

**Table 4 animals-13-00586-t004:** Effect of TA on cecal fermentation of Brandt’s voles.

	Control	LT	HT
Ruminal SCFAs (μmol/g)			
Acetic acid	51.01 ± 1.66 ^a^	45.17 ± 1.22 ^b^	44.57 ± 1.99 ^ab^
Propionic acid	27.88 ± 2.03	20.56 ± 2.54	21.71 ± 2.61
Butyric acid	25.61 ± 2.70 ^a^	23.38 ± 1.58 ^ab^	17.02 ± 1.85 ^b^
Ruminal pH	7.33 ± 0.09	7.64 ± 0.08	7.36 ± 0.08
Ruminal ammonia N (μg)	1.68 ± 0.05	1.36 ± 0.09	1.30 ± 0.13

^a, b^ Means in the same row with different letters differ significantly (*p* < 0.05). Control, 0 mg•kg^−1^d^−1^ TA; LT, 600 mg•kg^−1^d^−1^ TA; HT, 1200 mg•kg^−1^d^−1^ TA; SCFAs, short-chain fatty acids. Data are presented as mean ± SEM, *n* = 8.

**Table 5 animals-13-00586-t005:** Effect of TA on the colonic microbial α-diversity of Brandt’s voles.

	Control	LT	HT
Feature	544.67 ± 8.66	543.00 ± 5.36	506.00 ± 31.61
Ace	574.51 ± 8.44	570.00 ± 7.60	540.83 ± 23.57
Chao1	582.19 ± 10.80	574.33 ± 7.90	545.68 ± 25.01
Simpson	0.97 ± 0.01	0.98 ± 0.00	0.97 ± 0.01
Shannon	6.70 ± 0.13	6.71 ± 0.10	6.52 ± 0.27

Control, 0 mg•kg^−1^d^−1^ TA; LT, 600 mg•kg^−1^d^−1^ TA; HT, 1200 mg•kg^−1^d^−1^ TA. Data are presented as mean ± SEM, *n* = 6.

**Table 6 animals-13-00586-t006:** KEGG pathways with significantly different abundances in the colonic microbiota of Brandt’s voles.

KEGG Pathways	*p*-Value
Ubiquinone and other terpenoid-quinone biosynthesis (Metabolism)	0.031
Steroid hormone biosynthesis (Metabolism)	0.032
Other glycan degradation (Metabolism)	0.024
Lipopolysaccharide biosynthesis (Metabolism)	0.010
Glycosphingolipid biosynthesis—ganglio series (Metabolism)	0.040
Ethylbenzene degradation (Metabolism)	0.049
Biosynthesis of siderophore group nonribosomal peptides (Metabolism)	0.032
Bacterial chemotaxis (Cellular Processes)	0.030
Flagellar assembly (Cellular Processes)	0.021
Apoptosis—fly (Cellular Processes)	0.026
Lysosome (Cellular Processes)	0.039
Apoptosis (Cellular Processes)	0.013
PPAR signaling pathway (Organismal Systems)	0.045
Adipocytokine signaling pathway (Organismal Systems)	0.009
Protein digestion and absorption (Organismal Systems)	0.046
Salmonella infection (Human Diseases)	0.022
Pertussis (Human Diseases)	0.031
Staphylococcus aureus infection (Human Diseases)	0.014

## Data Availability

The corresponding author can provide the data described in this study upon request.

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
