# Peer review of "Tannic Acid Induces Intestinal Dysfunction and Intestinal Microbial Dysregulation in Brandt’s Voles (Lasiopodomys brandtii)"

_animals, 2023, doi:10.3390/ani13040586_

Round 1
Reviewer 1 Report
I have also some concerns regarding the manuscript grammar and sentence structure that is given below
Simple summary
Sentence is not clear ‘Long-term evolution has resulted in plants developing a defense mechanism against herbivores, which interferes with animal eating, digestion, and metabolism, thus restricting animals' food choices’ please rewrite it
Abstract
Please add p values of your results in abstract section
This sentence ‘This study was aimed at investigating the effects of dietary tannic acid (TA) on digestion and mucosal barrier function in Brandt's voles’ is not clearly reflecting your study. Please write your full objective here.
Don’t start the sentence with abbreviation ‘TA also weakened the digestive ability by reducing digestive enzyme activity’
Don’t start the sentence with abbreviation ‘TA significantly changed the intestinal permeability, as evidenced by an increase in serum zonulin levels’
Don’t start the sentence with abbreviation ‘TA reduced the numbers of intestinal goblet cells, and mast cells and the levels of serum cytokines and immunoglobulins (e.g., IgA, IgG, TNF-α, IL-6 and duodenal SlgA)’
Your conclusion is not based on your all results, please include all results ‘n conclusion, the results indicate that TA could damage the intestinal function of Brandt's voles’
secondary metabolite have already abbreviated in second sentence of abstract. Please use abbreviation instead of full term ‘Our study initially investigated the effects of natural plant secondary metabolites on the intestinal function of wildlife, and improved our general understanding of plant-herbivore interactions and the ecological role of PSMs’
Introduction
Reference is missing ‘Tannic acid (TA), a kind of hydrolyzable tannins, alters the growth performance of animals’
Broilers and pigs are not herbivores, please cite appropriate reference for your justification ‘Tannins could induce intestinal mucosal necrosis and villus damage in broilers…………….’
Your study include ‘ (1) growth performance, (2) the intestinal morphology of the duodenum, jejunum, and ileum, (3) the activity of di[1]gestive enzymes and cecal fermentation, including the pH of the cecum together with the contents of short-chain fatty acids (SCFAs) and ammonia nitrogen (NH3-N), (4) redox status of the small intestine, and (5) immune and microbial barrier functions of the intestine’ which is not supported by the actual referencing data of introduction. Please include all review related to your manuscript in the introduction section
Materials and Methods
What do you mean by used? In this study? If yes please clarify in your sentence ‘Twenty-four male 13-week-old Brandt's voles, weighing 52.16 ± 0.84 g, were used’
The data of body weight, and feed intake could not measure in groups for statistical analysis. how could you defend it
Discussion
What do you mean by ‘The differences between studies might be due to difference in the amount of tannins and the duration of the experiments’ please justify your results
Reviewer 2 Report
The abstract is complete and well-written.
The introduction has properly covered all aspects involved.
Discussion section is complete and the findings are well established.
Reviewer 3 Report
The manuscript is clearly described and supported by recent bibliography, the statistical analysis is detailed as are the other methods used. I consider this work suitable for publication.
Reviewer 4 Report
This study found that the addition of tannic acid to feeds produced important effects on growth performance, intestinal morphology, digestive enzyme activity, cecum fermentation, intestinal barrier function, and intestinal flora in Brandt's voles. This study is interesting, has a good experimental design, and improves our broad understanding of plant-herbivore interactions. These following suggestions could better improve the quality of the article.
Introduction:
1. In line 4, would it be more accurate to add "some" in front of structural features? There are some spelling and grammar errors in the article, please check yourself in full.
Materials and methods:
2. The introduction section mentioned that there was an effect on the ovaries of mice, so why were only male mice selected for the experimental section? Are there any differences in the age of different sexes?
3. What is the reference basis for tannic acid dose setting? Why not set three doses of high, medium and low?
4. The overall line is not standardized enough, suggest using standard English for expression, such as"Baby weight" should be changed to" Body weight".
5. What is the blood collection method? Is there any mixed clotting?
6. What is the concentration of sucrose?
7. The experimental steps are incomplete. Should there be at least an embedding process? Or you can cite the steps in the references, but you need to explain.
Results:
8. The results section does not clearly state which dose of tannic acid has the greatest effect on the growth performance of herbivores, or whether it is dose-dependent?
9. The result part of the article is mostly shown in the form of trilinear table, can it be shown in the form of bar chart or other icons to enhance the readability of the article?
10. Weighing of internal organs can not be used such as, those internal organs weighed please write out clearly.
Discussion:
11. Page 11, line 26, "In vitro" is spelled incorrectly,and confirm if italics are needed.
Round 2
Reviewer 1 Report
In the current study twenty-four male 13-week-old Brandt's voles, weighing 52.16 ± 0.84 g, were used in three groups of eight voles each which could not justify the results especially of feed intake and body weight gain. Individually feeding is required, keeping in view the parameters of current study, therefore, I don’t believe this manuscript should be published
Author Response
We are sorry that the details were not clearly stated. In our experiment, we guaranteed that each vole was housed and raised individually, which we have also explained in the manuscript. We gratefully appreciate for your valuable comment.